# Aromatic Diboronic Acids as Effective KPC/AmpC Inhibitors

**DOI:** 10.3390/molecules28217362

**Published:** 2023-10-31

**Authors:** Joanna Krajewska, Piotr Chyży, Krzysztof Durka, Patrycja Wińska, Krystiana A. Krzyśko, Sergiusz Luliński, Agnieszka E. Laudy

**Affiliations:** 1Department of Pharmaceutical Microbiology and Bioanalysis, Medical University of Warsaw, 02-097 Warsaw, Poland; joanna.krajewska@ymail.com; 2Centre of New Technologies, University of Warsaw, 02-097 Warsaw, Poland; p.chyzy@cent.uw.edu.pl; 3Faculty of Chemistry, Warsaw University of Technology, 00-664 Warsaw, Poland; kdurka@gmail.com (K.D.); pwinska@ch.pw.edu.pl (P.W.); sergiusz.lulinski@pw.edu.pl (S.L.); 4Faculty of Physics, University of Warsaw, 02-093 Warsaw, Poland

**Keywords:** arylboronic acids, KPC/AmpC β-lactamase inhibitors, molecular docking, time-dependent QM/MM, antibacterial activity

## Abstract

Over 30 compounds, including *para*-, *meta*-, and *ortho*-phenylenediboronic acids, *ortho*-substituted phenylboronic acids, benzenetriboronic acids, di- and triboronated thiophenes, and pyridine derivatives were investigated as potential β-lactamase inhibitors. The highest activity against KPC-type carbapenemases was found for *ortho*-phenylenediboronic acid **3a**, which at the concentration of 8/4 mg/L reduced carbapenems’ MICs up to 16/8-fold, respectively. Checkerboard assays revealed strong synergy between carbapenems and **3a** with the fractional inhibitory concentrations indices of 0.1–0.32. The nitrocefin hydrolysis test and the whole cell assay with *E. coli* DH5α transformant carrying *bla*_KPC-3_ proved KPC enzyme being its molecular target. *para*-Phenylenediboronic acids efficiently potentiated carbapenems against KPC-producers and ceftazidime against AmpC-producers, whereas *meta*-phenylenediboronic acids enhanced only ceftazidime activity against the latter ones. Finally, the statistical analysis confirmed that *ortho*-phenylenediboronic acids act synergistically with carbapenems significantly stronger than other groups. Since the obtained phenylenediboronic compounds are not toxic to MRC-5 human fibroblasts at the tested concentrations, they can be considered promising scaffolds for the future development of novel KPC/AmpC inhibitors. The complexation of KPC-2 with the most representative isomeric phenylenediboronic acids **1a**, **2a**, and **3a** was modeled by quantum mechanics/molecular mechanics calculations. Compound **3a** reached the most effective configuration enabling covalent binding to the catalytic Ser70 residue.

## 1. Introduction

Aromatic di- and triboronic acids are popular building blocks used in organic synthesis, for the construction of extended hydrogen-bonded supramolecular assemblies [1] and porous materials with a special emphasis on Boronate Covalent Organic Frameworks [2]. With a few exceptions [3], however, their potential in medicinal chemistry has not been extensively investigated, yet. In this context, it is worth noting that the presence of two or more boronic groups attached to the aromatic backbone results in reduced lipophilicity and increased acidity compared to monoboronic derivatives which are more popular in medicinal chemistry, e.g., they were recognized as potent β-lactamase inhibitors (BLIs) [4]. β-Lactamases, as enzymes hydrolyzing β-lactams, are the main cause of Gram-negative rod resistance to these antibiotics. This mechanism of bacterial resistance has a negative impact on global public health [5,6,7,8,9,10]. Moreover, from the point of medicinal chemistry, boronic acids and their derivatives have also found other applications, e.g., they have emerged as covalently binding proteasome inhibitors and the most important example is bortezomib used as the anticancer agent for the treatment of multiple myeloma [11].

Among β-lactamases, carbapenemases are the most troublesome as they extend bacterial resistance to the last-resort antibiotics—carbapenems [5]. The most clinically relevant carbapenemases belong mainly to class A according to Ambler classification [6], (e.g., *Klebsiella pneumoniae* carbapenemases—KPCs), class B (the so-called metallo-β-lactamases—MBLs, e.g., New Delhi metallo-β-lactamases—NDMs, Verona integron-encoded metallo-β-lactamases—VIMs), and class D (e.g., carbapenem-hydrolyzing class D β-lactamases—CHDLs and enzyme OXA-48). Class C carbapenemases were rarely described, e.g., ADC-68 in *Acinetobacter baumannii* [5,7]. However, the widespread resistance to third and fourth-generation cephalosporins among bacteria is also worrisome [8]. It results either from the production of the class A extended-spectrum β-lactamases (ESBLs, e.g., CTX-M enzymes) or from the production of class C cephalosporinases (AmpC), both chromosomal (cAmpC) and plasmid-encoded (pAmpC, e.g., CMY-2) [8,9].

Considering the above, in 2017, the World Health Organization (WHO) classified carbapenem-resistant strains of *A. baumannii* (CRAB), *Pseudomonas aeruginosa* (CRPA), and *Enterobacterales* (CRE) as well as third-generation cephalosporin-resistant *Enterobacterales* as the “critical priority pathogens”, for which therapeutic options are severely limited [10]. However, the number of new antibacterial agents, both recently approved and under clinical and preclinical development, is still insufficient to address this problem [12,13]. Combining β-lactams with BLIs is a well-known strategy to restore the effectiveness of these antibiotics. BLIs based on the β-lactam scaffold (clavulanic acid, sulbactam, and tazobactam) entered clinics in the 1980s–1990s [14] but they are inactive against carbapenemases. Recently, market authorization gained three non-β-lactam inhibitors of class A carbapenemases (avibactam, relebactam, and vaborbactam) and one potent inhibitor of class A and D carbapenemases (durlobactam) [15,16]. However, class B carbapenemases are still out of the spectrum of clinically available BLIs. Thus, searching for new broad-spectrum inhibitors is urgently needed [13,15].

Boronic acids and their derivatives are well-known groups of competitive, reversible BLIs [4,17]. They react with the nucleophilic serine in the catalytic center of β-lactamases, forming tetrahedral complexes [4]. Vaborbactam (Figure 1), based on cyclic boronate (boronic acid monoester) pharmacophore, was already marketed in combination with meropenem for the treatment of complicated urinary tract infections. It is a potent inhibitor of class A carbapenemases (KPC, SME) and ESBLs (CTX-M, TEM, and SHV) as well as AmpC enzymes [12]. Three cyclic boronates with the activity extended toward many metallo-β-lactamases and class D enzymes are under clinical trials (taniborbactam, VNRX-7145, and QPX7728) [13] whereas other derivatives are under preclinical development [18]. (2′-*S*)-(1-(3′-Mercapto-2′-methylpropanamido)methyl)boronic acid was recently found to inhibit a broad spectrum of β-lactamases, including some MBLs (VIM-2 and NDM-1) [19]. Various aromatic boronic acids (including phenylboronic acids) also display BLI activity against class A (KPC [20,21,22,23,24] and CTX-M [25,26]) and class C β-lactamases [23,26,27,28,29,30]). Improved affinity toward AmpC enzymes was found for many *meta*-substituted phenylboronic acids, e.g., those bearing amide [27], sulfonamide [27], aza-naphthol [29], and aza-phenol [29] moieties, while carboxyvinyl group in the *ortho* or 1,2,3-triazole in the *meta* position were associated with an increased KPC-2 inhibitory potency [22,23,24]. Finally, many benzoxaboroles manifest broad-spectrum in vitro inhibition of class A (TEM-1, CTX-M-15, and KPC-2), class C (cAmpC, CMY-2), and class D (OXA-10, OXA-24, and OXA-48) enzymes [17,31,32]. However, some compounds were not able to restore the effectiveness of antibiotics against strains producing particular β-lactamases [19,27,29,32]. This emphasizes the importance of performing microbiological whole-cell assays, to evaluate antibacterial activity. Considering the BLI activity of simple phenylboronic acid (PBA) [33], the major motivation of this work has been the practical evaluation of the effect of the introduction of additional boronic group(s) attached to the aromatic core using appropriate microbiological assays supported by theoretical modeling performed for the most representative systems.

## 2. Results and Discussion

### 2.1. Synthesis

Our study involved over 30 compounds including *para*-(**1a**–**1l**), *meta*-(**2a**–**2g**) and *ortho*-benzenediboronic acids **3a**–**3e** (Figure 2). In addition, benzenetriboronic acids (**4a**–**4b**) were the subject of our work. Further examples include di- or triboronated thiophene (**5a**–**5c**) and pyridine (**6a**) derivatives. Finally, a few *ortho*-substituted phenylboronic acids (**7a**–**7c**) featuring intramolecular hydrogen-bonding interactions were also investigated. Compounds **1a**, **2a** and **7a** were commercially available. The synthesis of most compounds including **1b**–**1g** [1,34,35], **1j**–**1l** [36,37], **2d**–**2g** [34,35], **3a**–**3d** [38], **4a**–**4b** [35], **5b**–**5c** [39], and **7b** [40] was reported previously. The synthesis of remaining boronic acids **1h**, **1i**, **2b**, **2c**, **3e**, **6a**, **7c** is presented in Figure 3. In general, they were obtained from appropriate aromatic precursors using the lithiation/boronation reaction sequence with some variations introduced to install the boronic groups effectively. The details of synthetic procedures and compound characterization are given in the Appendix A.

### 2.2. Direct Antimicrobial Activity

It should be emphasized that the main objective of this study was to evaluate the β-lactamase inhibitory capacity of the tested aromatic diboronic acids. However, comprehensive screening of their direct antimicrobial potency was also important. Many BLIs display intrinsic antibacterial activity via inhibiting penicillin-binding proteins (PBPs). It was reported for clavulanic acids [41], sulbactam [42], some diazabicylococtane BLIs [43,44,45,46], and recently for boronate BLI xeruborbactam [47]. Though the direct activity of BLIs is usually weak (the minimal inhibitory concentrations—MICs > 64 mg/L) [41,43,44,45] or moderate (e.g., 16 mg/L for xeruborbactam against CRE strains) [47], it is considered beneficial, as such compounds potentiate β-lactams also against non-β-lactamase-producing organisms. Among various organoboron compounds studied over the past decades [48], benzoxaboroles proved most effective as direct antimicrobials. They usually act as inhibitors of leucyl-tRNA synthetase (leuRS) in fungi [49,50], mycobacteria [51], Gram-positive [52,53], and Gram-negative bacteria [54,55]. Moreover, structurally similar benzosiloxaboroles displayed high activity against Gram-positive cocci (including methicillin-resistant *Staphylococcus aureus*—MRSA) [56] and yeasts [17]. However, the direct antimicrobial activity of some phenylboronic acids was also reported [57,58,59]. In this work, we screened the activity of all compounds against six Gram-positive strains, 10 Gram-negative strains, and six yeasts by disc diffusion method followed by the determination of MICs and minimal bactericidal/fungicidal concentrations (MBCs/MFCs). Obtained results are collected in Appendix A. A moderate-to-weak activity against Gram-positive strains was found for most tested compounds (Appendix A). The lowest MICs were obtained for compound **2e** (0.78–12.5 mg/L for staphylococci, including MRSA; 50–100 mg/L for enterococci) and for compound **7c,** which was equally potent against all tested coccis (MICs 12.5–25 mg/L). These results are comparable with our previous findings for various benzosiloxaboroles [56]. Besides, **1c, 1e**, **1j**, and **7b** displayed moderate activity against staphylococci (MICs 12.5–50 mg/L) and weak against enterococci (MICs 100–400 mg/L), whereas other compounds showed only weak activity regardless of the species. The weak activity (MICs 50–400 mg/L) against Gram-negative rods (except *P. aeruginosa*) was found for some *para- (***1b**–**1f**, **1i**, and **1j**) and *meta*-phenylenediboronic acids (**2b**–**2e**), as well as for **3a**, **7a**, and **7b** (Appendix A). Assuming that multidrug-resistant (MDR) efflux pumps may contribute to Gram-negative rod resistance, the activity of tested agents in the presence of the efflux pumps inhibitor phenylalanine-arginine-β-naphthylamide (PAβN) was also examined [60,61,62]. Interestingly, only MICs of compounds **2e**, **3e**, and **7b** were significantly reduced in the presence of PAβN (by 4–16-fold). Therefore, it is unlikely that the activity of other compounds is affected by the efflux phenomenon. This is in contrast to benzosiloxaboroles which were actively extruded from bacterial cells [17]. Most tested agents displayed only weak activity against reference yeasts (Appendix A). Exceptionally, **3e** was highly active against most *Candida* spp. (MICs 3.12–6.25 mg/L) and moderately active against *Candida krusei* (MIC 25 mg/L) and *Saccharomyces cerevisiae* (MIC 50 mg/L). These values are comparable with those we reported for some benzosiloxaboroles [17,56] and lower than those obtained for various phenylboronic acids [57,58,63]. Moreover, moderate antifungal activity was found for **1i** and **7a**–**7c** (MICs 6.25–50 mg/L). The fungicidal activity was noticed in the case of **3e** (against *Candida albicans*, *Candida tropicalis,* and *Candida guilliermondii* with the MFCs 400 mg/L) and **7c** (against *C. albicans*, *C. tropicalis, C. guilliermondii,* and *S. cerevisiae* with the MFCs 25–200 mg/L). To summarize, most of the tested aromatic boronic acids displayed rather weak activity against Gram-positive strains whereas only a few derivatives inhibited the growth of Gram-negative rods and yeasts.

### 2.3. BLI Activity at High Concentrations

The BLI activity of many boronic acids [4] including some derivatives reported by us recently [17] has prompted us to investigate all of the presented compounds. First, we performed three combination disc tests (CDTs) searching for KPC, AmpC, and ESBL inhibition at high concentrations of tested compounds. PBA was used as the reference inhibitor of both KPC [64] and AmpC [65] enzymes at a concentration of 0.3 mg per disc. Considering that some tested agents inhibited the growth of reference Gram-negative rods, screening tests of direct antimicrobial activity (STDA) against β-lactamase producers were performed using the disc-diffusion method. Discs with 0.3, 0.1, and 0.03 mg of each agent were examined. In the CDTs, concentrations lower than 0.3 mg per disc were used in the case of **1b**–**1f**, **1i**, **1j**, **2c**, **2d**, **7a**, and **7b** (Appendix A). A significant difference in the growth inhibition zone diameters around the antibiotic disc with an agent *versus* the same antibiotic disc without an agent was taken as an indicator of BLI activity (Table 1) [64,65]. BLI activity against at least one tested strain was found for the majority (25 out of 33) of tested compounds. Three phenylenediboronic acid regioisomers (**1a, 2a, 3a**), their fluorinated derivatives (**1b**, **1d**, **1e**, **2b**, **3b**, **3c**), benzene-1,2,4-triboronic **4b**, thiophene-2,3-diboronic acid **5a** and 2,5-difluorophenylboronic acid **7a** increased the meropenem activity against the KPC-2-positive strain and ceftazidime activity against both strains producing cephalosporinases (chromosomally encoded cAmpC and plasmid-encoded CMY-2). Some functionalized phenylenediboronic acids, i.e., those bearing fluorine (**1c**, **1f**, **2c**, **2d**, **3d**), bromine (**1i**, **2f**), CF_3_ (**2e**, **3e**), or OMe (**2g**) substituents increased the sensitivity of AmpC-producers only.

The same activity profile was found for benzene-1,3,5-triboronic acid **4a** and thiophene-2,3,5-triboronic acid **5c**, whereas 2-mercaptophenylboronic acid **7c** increased only the sensitivity of *K. pneumoniae* KPC-2-positive. Lack of BLI activity was observed for phenylene-1,4-diboronic acids including perfluoro derivative **1g**, and compounds bearing bulkier substituents, i.e., CF_3_ (**1h)**, I, SiMe_3_, OMe, (**1j**–**1l**, respectively). Two heteroaromatic compounds, namely thiophene-2,5-(**5b**) and pyridine-3,5-diboronic acid hydrochloride (**6a**) were also inactive. It should be noted that none of the tested boronic acids showed any activity against ESBL-positive *K. pneumoniae*.

### 2.4. BLI Activity at Low Concentrations

Subsequently, we performed microdilution tests for 25 agents, which displayed BLI activity in the CDTs. They were used at low concentrations (16, 8, and 4 mg/L) as reported previously for known BLIs (vaborbactam, relebactam, and avibactam) [66]. Under these conditions, 18 compounds are inactive against tested Gram-negative bacilli (MICs ≥ 400 mg/L), whilst seven present only weak direct activity (MICs ranging from 100–200 mg/L, Appendix A). First, we evaluated the capabilities of reducing ceftazidime MICs for two AmpC producers as well as meropenem MICs for four KPC producers and one VIM-positive strain. All KPC- and VIM-positive strains used at this stage were resistant to tested carbapenems, whereas all AmpC-positive strains were resistant to ceftazidime according to EUCAST breakpoints [66] (Appendix A). PBA was used as a reference inhibitor of both KPC [64] and AmpC [65] enzymes. At least the 4-fold reduction in a β-lactam MIC was taken as an indicator of BLI activity. Also, the ability of tested agents to restore susceptibility (i.e., to reduce antibiotic MIC to or below EUCAST breakpoint [66]) was examined. Seventeen compounds displayed BLI activity against KPC or AmpC producers. In contrast, none of the tested agents increased the susceptibility of *P. aeruginosa* VIM-positive to meropenem. **1a**, **2a**, **3a**, their fluorinated derivatives (**1b**–**1f**, **2b**–**2d**, and **3b**), **4b**, **5a**, **5c**, and **7a** displayed BLI activity towards both cAmpC- and CMY-2-positive strains (Appendix A). The strongest ceftazidime MIC reductions were obtained in the presence of *para*-phenylenediboronic acids **1a**–**1f** and **7a** (up to 32/16/8-fold at 16/8/4 mg/L of a tested agent, respectively), slightly smaller in the presence of *meta*-phenylenediboronic acids **2a**–**2d** (up to 16/8/4-fold at 16/8/4 mg/L, respectively). All active *para*- and *meta*-phenylenediboronic acids and **7a** reduced ceftazidime MIC of *Escherichia coli* CMY-2-positive to ≤8 mg/L, thus reaching EUCAST breakpoint for ceftazidime in the presence of avibactam (at 4 mg/L) [66]. However, only **1a** and **7a** resensitized *P. aeruginosa* cAmpC-positive to ceftazidime (breakpoint in the presence of avibactam at 4 mg/L also equal 8 mg/L [66]). Compounds **3a**, **3b**, **4b**, **5a**, and **5c** were less potent (MIC reductions up to 8/4/2-fold at 16/8/4 mg/L, respectively), and they did not restore sensitivity to ceftazidime of either of the AmpC-positive strains. The potency of *para*- and *meta*-phenylenediboronic acids and **7a** in increasing sensitivity of *E. coli* CMY-2-positive was higher than that for PBA, but weaker in the case of *P. aeruginosa* cAmpC-positive. Compared to unsubstituted acids **1a** and **2a**, their fluoro derivatives were generally more active towards *E. coli* CMY-2-positive (up to 4-fold for **1e**) but less active towards *P. aeruginosa* cAmpC-positive (also up to 4-fold).

BLI activity toward KPC producers was less common than toward AmpC producers. However, significant reductions of meropenem MIC of at least two KPC-positive strains were obtained for 7 compounds: **1a** and its fluorinated derivatives (**1b**, **1d**, **1e**), **3a** and its 4-fluoro derivative **3c**, and **7a** (Appendix A). We suppose that the lack of activity of *meta*-phenylenediboronic acids against KPC producers can be ascribed to different structures of binding sites of KPC and AmpC enzymes. Thus, it seems that the mutual *meta* orientation of two boronic groups (the case for **2a** and its derivatives) disfavors binding to KPC. The active agents were subsequently tested with other carbapenems, i.e., with imipenem and ertapenem (Table 2). Among the total of 84 combinations with each carbapenem, significant MIC reductions were obtained 53 times for imipenem, 48 times for meropenem, and 41 times for ertapenem. Each tested agent potentiated all three carbapenems comparably (MIC reductions in most cases differ no more than by 2-fold, rarely by 4-fold).

The highest activity was found for **3a** (carbapenems’ MIC reduction up to 64/16/8-fold at 16/8/4 mg/L of tested agent, respectively). Compounds **1a** and **7a** were less potent (MIC reduction up to 16/8/2-fold at 16/8/4 mg/L, respectively). For comparison, PBA at 16 mg/L reduced carbapenems’ MICs only up to 8-fold for KPC-2 producers and up to 4-fold for KPC-3 producers. All seven agents and PBA reduced meropenem MIC of KPC-3-positive strains to ≤8 mg/L—*Enterobacterales* breakpoint for meropenem in the presence of vaborbactam (at 8 mg/L) [66] (Table 2). Six agents (all except **3c**) also resensitized *K. pneumoniae* ATCC BAA-1705 KPC-2-positive, whereas **1a** and **3a** restored the meropenem activity against all studied KPC producers. Moreover, all seven agents reduced imipenem MIC for *K. pneumoniae* 81 KPC-3-positive to ≤2 mg/L (breakpoint in the presence of relebactam at 4 mg/L [66]). Compounds **1a**, **1b**, **3a**, **3c**, and **7a** (unlike PBA) resensitized *K. pneumoniae* 83 KPC-3/CTX-M-3-positive to imipenem, whereas **3a** also resensitized *K. pneumoniae* KPC-2-positive. We did not achieve a reduction of ertapenem MIC to its susceptibility breakpoint (0.5 mg/L, used alone as this antibiotic is not combined with any BLI [66]). However, it is worth noting that **1a** and **3a** potentiated ertapenem better than PBA toward all KPC producers (by up to 8- and 16-fold, respectively). Recently, α-amido-β-triazolylethaneboronic acid at 4 mg/L was found to restore the susceptibility of strains producing various KPC, SHV, TEM, and CTX-M enzymes to ertapenem. However, it should be noted that strains used in our study were much more resistant to this carbapenem (MIC ranges 32–256 mg/L vs. 4–16 mg/L in the previous report [20]).

KPC-2 and KPC-3 enzymes are the most prevalent serine carbapenemases, which differ only in the amino acid at position 272 (histidine in KPC-2, tyrosine in KPC-3) [5]. Interestingly, tested boronic acids potentiated carbapenems better against *K. pneumoniae* 81 KPC-3-positive than against KPC-2 producers (2–4-fold higher MIC reductions). Thus, in order to clearly indicate the molecular target of tested agents, the most representative derivatives **1a**, **2a**, **3a**, and reference PBA were additionally tested alone and in combination with meropenem against meropenem susceptible *E. coli* DH5α and against *E. coli* 82 TR(pl 81)—the transformant of *E. coli* DH5α carrying a plasmid with *bla*_KPC-3_ gene from the clinical strain of *K. pneumoniae* 81 KPC-3-positive. Tested agents alone were inactive against both the parent strain and its transformant (MICs of PBA, **1a**, and **2a** > 400 mg/L, MICs of **3a** equal to 400 mg/L). None of them (in concentrations of 4, 8 and 16 mg/L) altered meropenem MIC of *E. coli* DH5α (Table 3). In contrast, significant (at least 4-fold) meropenem MIC reductions were obtained in their presence for *E. coli* 82 TR(pl 81). As expected, **3a** turned out to be the most potent compound, reducing meropenem MIC by 32/16/4-fold at the concentration of 16/8/4 mg/L, respectively. PBA and **1a** were less effective, whereas **2a** reduced meropenem MIC only by 2-fold. Subsequently, total proteins from *E. coli* 82 TR(pl 81) cells were extracted and the nitrocefin hydrolysis test was performed. KPC-3 was the only β-lactamase produced by *E. coli* 82 TR(pl 81). Efficient nitrocefin hydrolysis in the positive control measurement as relative absorbance proved the presence of KPC-3 in the purified protein extract. (Appendix A). The reduction in the relative absorbance level by 53/44/42% in the presence of 16 mg/L, and by 43/35/35% in the presence of 8 mg/L of PBA/**3a**/**1a**, respectively, indicated tested agents as effective KPC-3 inhibitors (Table 3 and Appendix A). Compound **2a** showed only weak BLI activity in concentration 16 mg/L, 29% reduction in relative absorbance. Interestingly, PBA turned out to be the most effective KPC-3 inhibitor in the nitrocefin hydrolysis test, despite its inferiority in potentiating carbapenems against *E. coli* 82 TR(pl 81) as well as clinical KPC producers. Thus, the physicochemical properties of the tested diboronic acids enable their better performance in microbiological whole-cell assays.

Overall, more agents were active against strains producing cephalosporinases than carbapenemases (16 vs. 7 compounds) (Appendix A). However, six compounds (**1a**, **1b**, **1d**, **1e**, **3a** and **7a**) displayed BLI activity toward both KPC and AmpC producers. Interestingly, apart from parent aromatic diboronic acids **1a**, **2a**, and **3a**, only their fluorinated derivatives (except for **1c**) were effective BLIs toward KPC producers. Moreover, 13 compounds restored the sensitivity of at least one strain (Appendix A). Notably, **3a** was the most potent agent in resensitizing KPC producers to meropenem (all four strains) and imipenem (3 strains). However, it did not restore the sensitivity of any AmpC producer to ceftazidime. Compound **1a** restored the sensitivity of all KPC producers to meropenem, all KPC-3 producers to imipenem, and two AmpC-positive strains to ceftazidime (Table 2). Fluorinated derivatives **1b**, **1d**, and **1e** proved slightly less successful as the meropenem breakpoint for *E. coli* KPC-2-positive and the ceftazidime breakpoint for *P. aeruginosa* cAmpC-positive were not reached, whereas imipenem breakpoint for *K. pneumoniae* KPC-3/CTX-M-3-positive was not reached with **1d** and **1e**. Compound **7a** was comparably active, as only the meropenem breakpoint for *E. coli* KPC-2-positive was not reached in its presence.

### 2.5. Synergy Evaluation

To investigate the type of interaction between studied boronic acids and β-lactams, we performed checkerboard assays and calculated fractional inhibitory concentrations indices (FICIs) for compounds that caused at least one significant reduction in any β-lactam MIC. This part of the work included 16 compounds, and PBA as a reference, combined with ceftazidime against two AmpC producers and seven tested agents, and reference PBA, combined with three carbapenems against four KPC producers. Following Bonapace et al., both the lowest and the average FICI were subsequently interpreted [67]. Obtained results are presented in Table 4 and Table 5.

Finally, assuming that the lower the average FICI (aFICI) value, the stronger the synergy, we compared aFICIs according to structural classification based on the mutual position and/or the number of boronic functionalities as well as the type of the aromatic ring. Thus, the tested agents were analyzed in the following groups denoted as **G1** (**1a**–**1f**), **G2** (**2a**–**2d**), **G3** (**3a**–**3b**), **G4** (**4b**), **G5** (**5a** and **5c**), and **G7** (PBA and **7a**) in the case of AmpC producers and as **GI** (**1a, 1b, 1d, 1e**), **GIII** (**3a** and **3c**), and **GVII** (PBA and **7a**) in the case of KPC producers. Additionally, we compared unsubstituted boronic acids denoted as **F1** (**1a**, **2a**, **3a**, **4a, 5a, 5c,** and PBA)/**FI** (**1a**, **3a**, PBA) with fluorinated derivatives **F2** (**1b−1f**, **2b−2d, 3b,** and **7a**)/**FII** (**1b**, **1d**, **1e**, **3c**, and **7a**) for AmpC/KPC producers, respectively. Tested strains were analyzed according to expressed β-lactamases. Due to a lack of normal distribution, the Kruskal–Wallis test was used, followed by Dunn’s multiple comparison tests if applicable. The significance level was set at *p* < 0.05. Obtained aFICIs for analyzed groups are presented in Figure 1 and Figure 2.

Synergy with ceftazidime (FICI ≤ 0.5) against both AmpC-positive strains was obtained for 14 agents, while synergy against one strain for the remaining two agents (**3a** and **3b**), regardless of whether the lowest or average FICI was interpreted (Table 4). Obtained FICIs are comparable with those recently reported for other phenylboronic acid derivatives combined with ceftazidime against *P. aeruginosa* AmpC-positive [23]. Subsequent statistical analysis revealed that tested combinations are comparably potent against both AmpC producers, as aFICIs did not differ significantly between *E. coli* CMY-2-positive and *P. aeruginosa* cAmpC-positive neither when all agents were analyzed together (*p_Kruskal–Wallis_* = 0.56), nor within each group separately (all *p_Kruskal–Wallis_* values > 0.05). However, the synergy between ceftazidime and **G3** was significantly weaker compared to **G2** (*p_Dunn_* = 0.04) and **G7** (*p_Dunn_* = 0.008) when both AmpC producers were analyzed together (Figure 1A). In turn, **F2** acted synergistically with ceftazidime significantly stronger than **F1** toward *E. coli* CMY-2-positive (*p_Kruskal–Wallis_* = 0.005). In contrast, differences between these groups were non-significant in the case of *P. aeruginosa* cAmpC-positive (*p_Kruskal–Wallis_* = 0.35) (Figure 1B).

Synergy with carbapenems was obtained for 65 per 84 cases, regardless of whether the lowest or average FICI was interpreted (Table 5). Obtained FICIs are comparable with those recently reported for some phenylboronic acids combined with meropenem against *K. pneumoniae* KPC-2-positive [23]. Interestingly, Celenza et al. previously reported that these combinations’ FICIs for strains with higher meropenem MICs are even lower [22]. The selected seven boronic acids potentiate each carbapenem comparably as aFICIs for their combination with imipenem, meropenem, and ertapenem did not differ significantly either when all KPC producers were analyzed together (*p_Kruskal–Wallis_* = 0.168), or within each β-lactamase group (all *p_Kruskal–Wallis_* > 0.05) (Figure 2B). It was confirmed that they potentiate carbapenems significantly better against the KPC-3-producing strain compared to the KPC-2 producer (*p_Dunn_* = 0.046) and KPC-3/CTX-M-3-positive one (*p_Dunn_* = 0.026). Moreover, significant differences in the strength of the synergistic interaction with carbapenems (Figure 2B,C) occurred among both groups (*p_Kruskal–Wallis_* < 0.0001) and **FI** vs. **FII** (*p_Kruskal–Wallis_* = 0.001). Regardless of the produced β-lactamase, **GIII** acted synergistically with carbapenems significantly stronger than **GI** (*p_Dunn_* values for KPC-2, KPC-3, and KPC-3/CTX-M-3 producers were 0.036, 0.007, and 0.006, respectively). In the case of the KPC-3-positive strain, their aFICIs with carbapenems were also significantly lower than the aFICIs of **GVII** (*p_Dunn_ =* 0.005). Moreover, synergy with carbapenems was significantly stronger for **FI** than for **FII** in the case of KPC-2 producers (*p_Kruskal–Wallis_ =* 0.0001) and KPC-3/CTX-M-3-positive strain (*p_Kruskal–Wallis_* = 0.033). This is in agreement with the recent findings by Zhou et al. who reported that fluoro derivatives of triazole-substituted phenylboronic acids are weaker KPC-2 inhibitors than unsubstituted compounds, even though fluorine substituents did not significantly alter the docked conformations [24].

Overall, the statistical analysis revealed that 16 arylboronic acids act synergistically with ceftazidime to a similar extent against CMY-2- and cAmpC-positive strains. In contrast, their interaction with carbapenems is significantly stronger against KPC-3- compared to KPC-2- and KPC-3/CTX-M-3-positive strains. The synergy is also comparable regardless of the carbapenem counterpart (imipenem, meropenem, ertapenem). However, synergy strength is influenced by both the structure variation and the presence of fluorine substituent(s). The synergy with carbapenems is the strongest for **GIII**, while synergy with ceftazidime is weaker for **G3** compared to **G2** and **G7**. Moreover, the installation of a fluorine substituent weakens synergy with carbapenems against KPC-2 producers and KPC-3/CTX-M-3 producers, simultaneously increasing the synergy with ceftazidime against CMY-2-positive strain.

### 2.6. Cytotoxicity Studies

The viability of MRC-5 human fibroblasts was tested after 72 h of treatment with each of the studied compounds used at the following concentrations: 12.5, 25, and 50 mg/L, except for **1a** and **3a** tested at 16, 32, and 64 mg/L. The obtained results are shown in Appendix A. Monoboronic acids **7b** and **7c** were the most cytotoxic with the viability of MRC-5 in the range of 0–56.4%. Other tested compounds decreased MRC-5 viability by no more than 50% and were less cytotoxic than PBA.

### 2.7. Molecular Modeling and Hybrid QM/MM Simulations

The crystal structure of a complex of 3-nitrophenylboronic acid (3-NPBA) with KPC-2, deposited in the Protein Data Bank (PDB id 3RXX [68]), was used as a starting point [17] to study the binding diboronic acids **1a**, **2a**, **3a**, and PBA as a reference compound. The active site of KPC-2 was described in the atomistic level of detail (Figure 3) [68,69]. It possesses S1 and S2 cavities (Figure 3A), surrounded by the Ω loop and loop between α3 and α4 helices, that are essential for competitive inhibition and ligand recognition [69,70] starting by anchoring the BLI in the S2 cavity. Critical amino acids (Lys73 and Glu166, see Figure 3B) are responsible for the formation of the protein-ligand dative covalent bond due to the reorganization of protonation states [69]. In turn, the S1 cavity comprises the catalytic serine (Ser70) with the O atom of the side hydroxymethyl group which is potentially able to bind to the sp^2^ hybridized boron atom of the BLI. Notably, the other crucial factors for the inhibition mechanism are also preserved, like Ser70 rotamer, the “flipped-out” position of Trp105 and the “in” position of Glu166 [68,69,70].

Molecular docking was performed for neutral and anionic forms of the chosen arylboronic acids. For **3a**, the structural specificity involving the formation of cyclic oxadiborole forms **3a_III**, **3a_IV**, and **3a_V** (Figure 4) was considered in accordance with the reported data [38]. For each ligand, we selected modes that fulfilled all criteria established for molecular docking evaluation. In general, most of the modes in the anionic form did not fulfill the distance criterion. The estimated free energies of binding do not significantly differ between ligands ranging from −4.48 ÷ −3.78 kcal/mol (Table 6). The difference of only 0.6 kcal/mol did not allow us to assess the ability of ligands for inhibition properly. For this reason, we performed molecular dynamics (MD) simulations. The selected modes achieved the thermodynamic equilibrium, which allowed us to assess their inhibitory potential.

We analyzed only the results of the stable fragment of the trajectory (time range from 1.8 to 2.0 ns). The most promising arrangement of each compound was selected based on the estimated average distance between the (Ser70)O and B atoms (*d*_Ser70(O)–B_, Table 6). If we performed MD simulations for various arrangements, we selected only those characterized by the lowest average values. By defining the cutoff set on 3 Å, we determined a set of arrangements able to form a protein-ligand covalent bond (see Appendix A).

MD simulations revealed that **2a** and **3a** forms bind to KPC-2 with similar interaction networks (Figure 4), but PBA and **1a** do not. It was caused by the steric clashes for **1a** (see Appendix A) and the lack of the second substituent in PBA, like in the NPBA. The interaction networks, shown in Figure 4, agree with the published experimental and computational studies [68,69].

The aromatic rings of **2a** and **3a** compounds occupy the same position as NPBA allowing it to interact with Trp105 via CH-π interactions (green dotted lines, Figure 4). In addition, such compounds form hydrogen bonds (HBs, blue dotted lines, Figure 4) that cover interactions published by Charzewski et al. [69]. The **2a** and all **3a** forms are stabilized by the HBs with the N atom of the Ser70 and Thr237 backbone, and the O atom of the Asn170 side chain (Figure 4), each with high occupancy (≥70%). In addition, the **3a_I** form creates an additional HB with the O atom of the Ser70 OH group. On the other hand, only in the MD simulations of **3a_IV** form, we detected a characteristic HB with the O atom of the Th237 OH group, observed in 99% of all analyzed simulation frames. It is worth noting that despite the same interaction network, the distance between the (Ser70) O and B atoms is significantly larger for the **2a** compound than for the **3a** (Table 6). The aromatic rings of PBA and **1a** are slightly shifted (compared to NPBA), limiting the hydrogen bonding to only the N atoms of the Ser70 and Thr237 backbone.

Finally, from the analyzed part of the trajectory, we extracted an arrangement with a minimal *d*_Ser70(O)–B_ value. We treated this arrangement as the most promising binding mode of each ligand. For such an arrangement (and a corresponding docking mode), we performed hybrid QM/MM simulations with eight repetitions (4 repetitions for each starting structure). For each diboronic acid, the formation of a protein-ligand covalent bond was observed, in agreement with the mechanism published by Charzewski et al. (Figure 5, Appendix A) [69]. The minimal time to form a protein-ligand covalent bond was estimated. PBA required 10.59 ps, 1a—8.49 ps, 2a—17.56 ps, **3a_I**—27.31 ps, **3a_III**—10.63 ps, and **3a_IV**—1.46 ps (Appendix A). Such a covalent bond was observed up to 150 ps, after which the simulations were stopped. It is qualitatively consistent with the reported experimental data (Table 2). The results indicate that all analyzed boronic acids can form a covalent bond with KPC-2 (Table 6 and Appendix A), and therefore, can be qualified as BLIs [69].

Based on the MD simulations of the most promising modes, the probability of favorable conditions for the formation of a covalent bond (P_NECESSARY_) was estimated. It can be quantified using the *d*_Ser70(O)–B_ value as a criterion. We concluded that such a rapprochement differs depending on the ligand. For PBA, P_NECESSARY_ was estimated as 40.4% of arrangements that are close enough which is the highest value among all analyzed ligands. The rest of the compounds have the following P_NECESSARY_: **1a**—2.6%, **2a**—1.2%, **3a_I**—11%, **3a_III**—22.4%, **3a_IV**—6.8%. This indicates that the mutual *ortho* location of boronic groups favors binding confirming the high potential of **3a**. However, apart from *d*_Ser70(O)–B_ value, other parameters such as the distance, angle, and active site amino acid conformations must also be optimal. Unfortunately, the impact of all these parameters cannot be predicted. We are able to find the correct distance and angle to form a covalent bond, but the random and unpredictable changes in the amino acid position do not allow us to determine the ideal conditions for nucleophilic addition. In our study, the use of crystallographic structure (with already defined Ser70, Glu166 and loops conformations enabling covalent binding) let us estimate the probabilities of covalent bond formation (P_SUFFICIENT_). This information we obtained from QM/MM simulations. Knowing the P_SUFFICIENT_, we can also estimate the probability of forming a covalent bond under favorable conditions (P_FAVORABLE_). Assuming that P_NECESSARY_ and P_SUFFICIENT_ are independent, P_FAVORABLE_ = P_NECESSARY_ × P_SUFFICIENT_. For PBA, P_FAVORABLE_ is 10.1%, for **1a**—0.65%, for **2a**—0.3%, for all forms **3a**—19.05% (**3a_I**—2.75%, **3a_III**—11.2%, **3a_IV**—5.1%). These results correspond to the experiments (Table 2) that allow us to relate higher P_FAVORABLE_ with lower MIC values, pointing out that the **3a** is the most active BLI.

It should be noted that the cyclic anionic form **3a_V** lacks the sp^2^-hybridized B atom needed to form a dative bond with Ser70 whilst simulations indicate that events involving the transformation of **3a_I** and **3a_III** into **3a_V** can occur (Appendix A). Interestingly, the transformation of **3a_III** required the participation of additional water molecules and Glu166 (Appendix A). Simulations also showed that the covalent docking process depends on the type of diboronic acid isomer. The binding mechanisms of **2a** and **3a** with the catalytic Ser70 residue start identically by filling the active site with an aromatic ring in the S2 cavity and forming the CH-π interaction with Trp105 (Figure 4). Next, the binding pose is relaxed, resulting in the stabilization via the HBs with the N atom of the Ser70 and Thr237 backbone, and the O atom of the Asn170 sidechain (Figure 4). Such an arrangement is waiting for a proton transfer from Lys73 to Glu166 and nucleophilic addition by the oxygen atom of the Ser70 OH group, leading to form a protein-ligand covalent bond. However, the larger average distance *d*_Ser70(O)–B_ for **2a**, suggests that the covalent bonding efficiency of BLI in *meta* substitution is lower than for *ortho*. For the **1a**, its *para* substitution prevents filling the narrow S1 and S2 cavities simultaneously forcing it to adopt a different docking process.

Notably, each form of **3a** differs in the stabilization process of an arrangement waiting for proton transfer. The **3a_I** creates a unique HB with the O atom of the Ser70 OH group. We speculate that this HB is unfavorable for the competitive inhibition mechanism. In **3a_I**, it reduces the number of beneficial arrangements of the catalytic Ser70 residue, ready for forming a protein-ligand covalent bond. In the case of the **3a_III**, the lack of the HB with the O atom of the Ser70 OH group enlarges the probability of favorable conditions for the formation of a covalent bond (Table 6). The most interesting is the **3a_IV** form, where the presence of an anionic form allows it to create the HB with the side chain of Thr237, which is an additional element in the stabilization process. Considering that the *ortho* substitution allows for cyclization, as well as that the anionic form creates an HB with the oxygen atom of the Thr237 OH group, we hypothesize that these factors allow the **3a** to be the most promising BLI.

## 3. Materials and Methods

### 3.1. Antimicrobial Activity

#### 3.1.1. Bacterial and Fungal Strains and Their Growth Conditions

Direct antimicrobial activity was determined in this study for the following strains: (1) Gram-negative bacteria from *Enterobacteriales* order: *Escherichia coli* ATCC 25922, *Klebsiella pneumoniae* ATCC 13883, *Proteus mirabilis* ATCC 12453, *Enterobacter cloacae* DSM 6234, *Serratia marcescens* ATCC 13880; (2) Gram-negative non-fermentative rods: *Pseudomonas aeruginosa* ATCC 27853, *Acinetobacter baumannii* ATCC 19606, *Stenotrophomonas maltophilia* ATCC 13637, *Burkholderia cepacia* ATCC 25416, *Bordetella bronchiseptica* ATCC 4617; (3) Gram-positive cocci: methicillin-sensitive *Staphylococcus aureus* ATCC 6538P (MSSA), methicillin-resistant *S. aureus* subsp. *aureus* ATCC 43300 (MRSA), *S. epidermidis* ATCC 12228, *Enterococcus faecalis* ATCC 29212, *E. faecium* ATCC 6057, *Bacillus subtilis* ATCC 6633; (4) yeasts: *Candida albicans* ATCC 90028, *C. parapsilosis* ATCC 22019, *C. tropicalis* IBA 171, *C. tropicalis* (Castellani) Berkhout ATCC 750, *C. guilliermondii* IBA 155, *C. krusei* ATCC 6258, and *Saccharomyces cerevisiae* ATCC 9763. The following strains were used for evaluating the BLI activity of tested agents: (1) two standard strains: *K. pneumoniae* ATCC BAA-1705 (with carbapenemase KPC-2) and *K. pneumoniae* ATCC 700603 (with extended-spectrum β-lactamase, ESBL, SHV-12); (2) six clinical isolates producing various classes of β-lactamases: carbapenemases KPC-2 (*E. coli* 76) and KPC-3 (*K. pneumoniae* 81 and 83), metallo-β-lactamase from VIM family (*P. aeruginosa* 1204), plasmid-acquired AmpC cephalosporinase CMY-2 (*E. coli* 77), and with overexpression of chromosomally encoded cephalosporinase AmpC (*P. aeruginosa* MUW 700); (3) *E. coli* DH5α and *E. coli* 82 TR(pl 81)—the transformant of *E. coli* DH5α with a plasmid from the clinical strain of *K. pneumoniae* 81 carrying the *bla*_KPC-3_ gene. All strains were stored at −80 °C. Prior to testing, each bacterial strain was subcultured twice on tryptic soy agar TSA (Biomaxima, Lublin, Poland) medium and yeast strains on Sabouraud dextrose agar (Biomaxima, Lublin, Poland) for 24–48 h at 30 °C to ensure viability.

#### 3.1.2. Determination of Direct Antimicrobial Activity

Direct antimicrobial activity against Gram-negative and Gram-positive bacterial strains, as well as against yeast, was examined as previously described [56] by the disc-diffusion test, the MIC determination assay, the MBC (for bacteria), and the MFC (for yeasts) determination tests. All the above-mentioned tests were performed according to EUCAST [71,72] and CLSI [73,74,75,76,77] recommendations. The following reference agents were used: nitrofurantoin (for Gram-negative rods), linezolid (for Gram-positive bacteria), and fluconazole (in the case of fungi). The new aromatic diboronic acids were dissolved in DMSO (Sigma, St. Louis, MO, USA). In the disc-diffusion test, the concentration of new agents was 0.4 mg per disc [17]. Depending on the solubility, the MIC and MBC/MFC values were determined up to 100 mg/L for **1k**, up to 200 mg/L for **7c**, and up to 400 mg/L for the remaining compounds: **1a**–**1j**, **2a**–**2g**, **3a**–**3f**, **4a**–**4b**, **5a**–**5c**, **6a** and **7a**–**7b**.

#### 3.1.3. Determination of MICs in the Presence of PAβN

To investigate the contribution of the MDR efflux pumps to the resistance of Gram-negative rods to the newly synthesized compounds, the MIC values of studied agents, with or without the pump inhibitor, PAβN (20 mg/L) (Sigma, St. Louis, MO, USA), were evaluated [78]. The MIC determination was performed in Mueller–Hinton II broth medium (MHB) (Becton, Dickinson and Company, Franklin Lakes, NJ, USA) using 2-fold serial dilutions of tested agents, according to the CLSI guidelines [75]. To minimize the influence of PAβN on the destabilization of bacterial cell covers, the tests were conducted in the presence of 1 mM MgSO_4_ (Sigma, St. Louis, MO, USA) [79]. At least a 4-fold reduction in the MIC value after the addition of PAβN was considered significant [80].

#### 3.1.4. Determination of BLI Activity

A two-stage approach was implemented for detecting the BLI activity of the tested agents. Initially, all compounds were subjected to combination disc tests (CDTs). For agents expressing BLI activity in the CDTs, microdilution tests were performed, and their synergy with antibiotics was evaluated.

##### Combination Disc Tests for Detection of BLI Activity

Prior to combination disc tests, screening tests of direct antimicrobial activity (STDA) of the tested agent against β-lactamase-producing strains were performed by the disc-diffusion method [72]. Discs with 0.3 mg, 0.1 mg, and 0.03 mg of each agent were examined. The highest amount of an agent that caused no effect on bacterial growth was used in further experiments. Concentrations that partially inhibited bacterial growth (isolated colonies or faint growth within the zone) were excluded from further experiments.

The three following CDTs were performed on Mueller–Hinton II agar (MHA) plates (Becton, Dickinson and Company, Franklin Lakes, NJ, USA) as described previously [17], according to the general EUCAST recommendations [64] and methodology described by Yagi et al. [65]. Briefly:CDT-KPC test for detection of KPC-type carbapenemase-producing strain was performed on the recommended strain *K. pneumoniae* ATCC BAA-1705. Discs with meropenem (MEM-10) (Becton, Dickinson and Company, Franklin Lakes, NJ, USA) alone and supplemented with one of the tested agents (TA) at the concentration consistent with the STDA result (0.3 mg or 0.1 mg or 0.03 mg per disc) were utilized. As the reference, the standard KPC inhibitor PBA (Sigma) at the concentration of 0.3 mg per disc was used. In this study, we considered the new compound has KPC-inhibitory activity when the increase in the diameter of the inhibition zone around MEM-TA vs. MEM-10 is at least 4 mm [64].CDT-AmpC test for detection class C β-lactamase-producing strain was performed on two clinical isolates: *P. aeruginosa* MUW 700 overexpressing chromosomally encoded cephalosporinase AmpC and *E. coli* 77 with plasmid-acquired AmpC cephalosporinase CMY-2. Discs with ceftazidime (CAZ-30) (Becton, Dickinson and Company, Franklin Lakes, NJ, USA), ceftazidime with a tested agent (CAZ-TA) at the concentration consistent with the STDA result, and ceftazidime with 0.3 mg of PBA (CAZ-PBA) as the reference AmpC inhibitor were utilized. We assumed the tested agent inhibits AmpC cephalosporinases when the diameter of the inhibition zone around CAZ-TA was at least 5 mm larger than that around CAZ-30 for both tested strains, considering that the same increase should be obtained for CAZ-PBA discs.CDT-ESBL EUCAST test for detection of ESBL-producing strain was performed on the recommended strain *K. pneumoniae* ATCC 700603, utilizing discs with ceftazidime (CAZ-30) (Becton, Dickinson and Company, Franklin Lakes, NJ, USA), ceftazidime with a tested agent (CAZ-TA) at the concentration consistent with the STDA result, and ceftazidime with 0.01 mg of clavulanic acid (CAZ-CL) (Becton, Dickinson and Company, Franklin Lakes, NJ, USA) as the reference ESBL inhibitor. In the case of clavulanic acid, the diameter of the inhibition zone around CAZ-CL should be at least 5 mm larger than that around CAZ-30 [64]. We consider the tested agent inhibits ESBL when the diameter of the inhibition zone around CAZ-TA is also at least 5 mm larger than that around CAZ-30.

##### Microdilution Tests for BLI Activity Detection and Synergy Evaluation

To examine the ability of tested agents to inhibit various carbapenemases and AmpC-enzymes, their synergy with antibiotics against β-lactamase-producing strains was assessed by checkerboard microdilution assay [67], with slight modification. Checkerboards were prepared in microtiter plates, with seven two-fold dilutions of β-lactams (1–64 mg/L) in the rows and three two-fold dilutions of tested agents and PBA (4–16 mg/L) in the columns. Tested agents’ concentrations were limited to the concentrations at which the newest commercially available BLIs (vaborbactam, relebactam, and avibactam) are used in susceptibility testing [66]. Plates were inoculated and incubated as in the MIC microdilution assay [75]. Parallel, MICs of tested agents and PBA alone were determined by the microdilution assay [75]. Following the incubation, antibiotic MICs in the presence of each agent’s concentration were determined. Moreover, for the first well without growth found in each checkerboard row and column along the growth/non-growth interface, the fractional inhibitory concentration index (FICI) was calculated using the formula below [67]:FICI = [(MIC of antibiotic in combination)/(MIC of antibiotic alone)] + [(MIC of the tested agent in combination)/(MIC of tested agent alone)].

For calculations, all off-scale MICs were converted to the next-highest doubling concentration. Subsequently, both the average and the lowest FICI were interpreted [67]. The interpretation was as follows: FICI  ≤  0.5, synergy; 0.5  <  FICI  ≤  4, indifference and FICI  >  4, antagonism [81]. Primarily, examined combinations consisted of meropenem (Pol-Aura, Morag, Poland) plus tested agents against carbapenemases-producing strains and ceftazidime (Pol-Aura, Morag, Poland) plus tested agents against AmpC-producing strains. For agents that reduced meropenem MIC at least 4-fold for at least two tested strains, similar assays were performed with imipenem (Pol-Aura, Morag, Poland) and ertapenem (Pol-Aura, Morag, Poland).

#### 3.1.5. Statistical Analysis

We analyzed aFICIs of compounds that caused at least one significant antibiotic MIC reduction. Owing to a lack of normal distribution, which was tested using the Shapiro–Wilk test, the analysis of variance (ANOVA) Kruskal–Wallis test was used to compare tested combinations’ average FICIs according to agents’ structural classification (6 groups of agents), agents’ substitution type (unsubstituted boronic acids vs. fluorinated derivatives), and carbapenem partner (for KPC producers). Post hoc analysis for Kruskal–Wallis ANOVA was conducted using a multiple comparison test (Dunn’s test). All statistical calculations were performed using STATISTICA version 13.3 PL (StatSoft, Cracow, Poland) software. The significance level was set at *p* < 0.05.

#### 3.1.6. Nitrocefin Hydrolysis Test

Overnight culture of *E. coli* 82 TR(pl 81) in MHB was diluted 1:100 into fresh MHB and incubated with shaking at 37 °C until an OD_600_ of 1 was attained. For the induction of the KPC-3 production meropenem was added to reach the concentration of 0.25 mg/L (0.125 × MIC). Further incubation was performed under the same conditions until an OD_600_ of 3 was reached. The culture with the required OD_600_ was centrifuged (6700× *g*, 10 min) and the supernatant was discarded. From the obtained bacterial cell pellet total proteins were extracted with the ReadyPreps™ Protein Preparation Kit (Epicentre Biotechnologies, Madison, WI, USA) and used in the subsequent nitrocefin hydrolysis test, performed in a 96-well microplate. Nitrocefin is a chromogenic cephalosporin substrate routinely used to detect the presence of β-lactamases. First, 20 μL of the protein extract was mixed with 80 μL of tested agents (**1a**, **2a**, **3a** and PBA as a reference BLI, each examined at concentrations: 4, 8 and 16 mg/L) or with 80 μL of the phosphate buffer (positive control—corresponding to KPC enzyme activity in the purified total proteins extracted from *E. coli* 82 TR(pl 81) cells). After 10 min of incubation at the room temperature 100 μL of the nitrocefin (Oxoid, Basingstoke, Hampshire, England) was added to reach its final concentration of 150 μM. β-lactamases hydrolyze the β-lactam ring of nitrocefin, causing its degradation and color change. Nitrocefin hydrolysis was evaluated after 3 min of the incubation at room temperature.

As in our previous publication [60] the presence of β-lactamases in the purified protein extract with and without tested agents was assessed by the spectrophotometric measurement of the rates of nitrocefin hydrolysis as relative absorbance at 486 nm. The level of the measured absorbance indicated β-lactamase activity. Finally, the difference in the relative absorbance between the positive control and the sample concentration of the tested inhibitors was calculated and expressed as a percentage. A reduction in the relative absorbance level in the presence of a tested agent was taken as an indicator of the BLI activity of a used aromatic diboronic acid. The experiment was performed in triplicate.

### 3.2. Cytotoxicity Studies

MRC-5 human fibroblasts (ECACC) were cultured in MEME, Minimum Essential Medium Eagle (Merck) supplemented with 10% fetal bovine serum (Merck), 2 mM l-glutamine, antibiotics (100 U/mL penicillin, 100 μg/mL streptomycin, Merck) and 1% non-essential amino acids (Merck). Cells were grown in 75 cm^2^ cell culture flasks (Sarstedt) in a humidified atmosphere of CO_2_/air (5/95%) at 37 °C. MTT-based viability assay was conducted as described previously [56]. Optical densities were measured at 570 nm using a BioTek microplate reader. All measurements were carried out in three replicates, and the results were expressed as a percent of viable cells versus control cells.

### 3.3. Docking and Time-Dependent Quantum Mechanics/Molecular Mechanics

#### 3.3.1. Structure Preparation and Molecular Docking

Computational studies were carried out to confirm the inhibitory properties of most representative derivatives of aromatic diboronic acids. As a reference system, the structure of carbapenemase KPC-2 (PDB ID: 3RXX [68]) was chosen, which proved the competitive inhibition mechanism of the BLI [17,25,69]. At first, the active site protonated at pH 7.0 was optimized [82] and minimized in the AMBER10 force field [83] using the Molecular Operating Environment (MOE, 2019) [84]. The semi-flexible docking protocol in the MOE was applied to predict the binding mode of each ligand (in the neutral and anionic form) [84]. As a result, ten modes of each ligand based on the lowest free energy of binding, estimated by the GBVI/WSA Δ*G* scoring function, were obtained [85]. A geometric analysis was made to distinguish the potential BLI by checking the distance between the oxygen atom of the serine hydroxyl group and the boron atom. The analysis involved only ligands with the above-mentioned distance below 4.5 Å, which allows the formation of the protein-ligand covalent bond. For the ligands with such ability, molecular dynamics calculations were performed to verify the binding stability.

#### 3.3.2. Molecular Dynamics Simulations

Each system was put in the rectangular simulation box, solvated with a 6Å water shell of TIP3P water molecules, and neutralized with NaCl ions. Then, we run the 2 ns molecular dynamics at the 310 K in the AMBER10:EHT force field [83] using the Molecular Operating Environment (MOE, 2019) [83]. The Nosé–Poincaré Andersen integrating algorithm was applied [85,86] with a 1.0 fs time step. The results of the most stable fragment of the entire trajectory (time range from 1.8 to 2.0 ns) were analyzed. Only arrangements that adopted the oxygen atom of the serine hydroxyl group-boron atom distance lower than 3 Å were selected. Based on them, the probability of favorable conditions (P_NECESSARY_) for the formation of a covalent bond was determined as P_NECESSARY_ = the number of frames with *d*_Ser70(O)-B_ ≤ 3Å divided by the number of frames in all the analyzed fragments of trajectory × 100%). If the multiple systems of the same ligand were obtained, the one with the best average estimation of the oxygen atom of the serine hydroxyl group-boron atom distance was picked. Finally, the arrangements with the lowest distance allowing for the formation of a protein-ligand covalent bond were determined. For these arrangements, QM/MM simulations were performed to validate the mechanism of the formation of protein-ligand covalent bonds.

#### 3.3.3. Quantum Mechanics/Molecular Mechanics

The performed hybrid approach employed a quantum mechanics component [69,87,88]. The MOPAC environment [89], integrated with NAMD software (version 2.12) [90] was used. The complexes were parameterized, prepared, and optimized in the CHARMM36 force field with the CGENFF parameters [91,92] using the NAMD/2.12 [90]. The published parameters and the simulation protocol [69]. Ligand atoms, selected amino acids of the active site (Ser70, Lys73, Ser130, Asn132, Glu166, Asn170, Lys234, Thr237), and the water molecules within 5 Å of any ligand atom were considered chemically important. Eight repetitions of the QM/MM, each including the 0.25 ps energy minimization and 50 ps simulation, were conducted. For the most promising arrangements, the simulation time was extended to 150 ps. VMD was used to detect the proton transfers and the formation of a protein-ligand covalent bond [93]. The probabilities of covalent bond formation (P_SUFFICIENT_) were estimated as a measure of sufficient conditions for forming a dative covalent bond (P_SUFFICIENT_ = number of the QM/MM repetitions with detected protein-ligand covalent bond divided by the number of all performed repetitions). The probability of the formation of a covalent bond under favorable conditions (P_FAVORABLE_), was estimated using the formula P_FAVORABLE_ = P_NECESSARY_ × P_SUFFICIENT_.

## 4. Conclusions

In conclusion, we found that many studied aromatic diboronic acids and their derivatives display potent BLI activity at low, clinically achievable concentrations. Respective SAR analysis for three series of diboronic acids **1a**–**1l**, **2a**–**2g**, **3a**–**3e** indicates that most of the fluorinated derivatives maintain activity comparable to respective parent compounds. In turn, the introduction of bulkier, lipophilic groups (I, CF_3_, SiMe_3_, OMe) has adverse effects and in general such compounds are not effective BLIs. Notably, selected agents are active against both KPC and AmpC enzymes responsible for the critical priority pathogen resistance. Moreover, they can restore the sensitivity of clinical strains to the last resort antibiotics (carbapenems, 3rd generation cephalosporins) at similar concentrations as inhibitors currently used in clinics. Among them, phenylene-1,2-diboronic acid **3a** was the most effective in potentiating carbapenems against KPC producers. This was confirmed by QM/MM simulations and the observed mechanism of the KPC-BLI covalent bond. Phenylene-1,3-diboronic acids potentiated ceftazidime against AmpC producers best, whereas phenylene-1,4-diboronic acids were highly effective in potentiating both carbapenems against KPC producers and ceftazidime against AmpC producers. Moreover, fluoro-substitution increased CMY-2 inhibitory activity, slightly reducing KPC/cAmpC inhibitory potency. Benzene-1,2,4-triboronic and boronated thiophenes increased only ceftazidime activity against AmpC producers to a moderate degree. Importantly, phenylenediboronic acids overcome simple PBA in KPC/AmpC inhibitory potency and, gratifyingly, display significantly reduced toxicity. Thus, it seems that the concept involving the introduction of the second boronic group to the structure can be considered a promising tool for the development of effective KPC/AmpC inhibitors. Since the functionalization with lipophilic groups seems to be ineffective, future work could involve the installation of substituents possessing a distinctive hydrophilic character, e.g., amide, amino acid, and peptide residues. We are planning to test this concept and the obtained results will be reported in due course.

## Data Availability

All obtained data in this work are included in the submitted manuscript.

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
