# Peer review of "Aromatic Diboronic Acids as Effective KPC/AmpC Inhibitors"

_molecules, 2023, doi:10.3390/molecules28217362_

Round 1
Reviewer 1 Report
Comments and Suggestions for Authors
Krajewska and co-authors describe in the manuscript “Aromatic diboronic acids as effective KPC/AmpC inhibitors” comprehensive investigations on the potential of aromatic diboronic acids as β-lactamase inhibitors (BLIs), including chemical synthesis, determination of direct antimicrobial activity, determination of synergy at high and low concentrations, cytotoxicity studies and molecular modelling based on structure from PDB. The investigations have been performed in a very systematic manner, and the analysis of the large set of data obtained is complete and insightful. This manuscript can be accepted for publication after minor revision.
Topics to be addressed in the revision:
1. The Introduction must be improved. Please describe the molecular mode of action of established boronate-type BLIs shortly and present structures of relevant compounds (vaborbactam, taniborbactam, 1-(3′-mercaptopropanamido)methyl)boronic acid).
Line 74/77: All of these BLIs are not “cyclic boronic acids”, but cyclic boronates (boronic acid monoesters).
Boronic acids are also used as covalently binding proteasome inhibitors (bortezomib and related drugs), this should be mentioned in the Introduction.
2. The authors mention previous work on aromatic boronic acids as BLIs (references 19-29, 32). Since this manuscript deals with aromatic boronic acids as well, the known SAR frpom the cited work must be concluded shortly.
3. Synthesis: Scheme 2: in the synthesis of 2b, step 2, B(OEt)3 was used, not B(OMe)3. Structure of pre-6a: present both boronate groups in a uniform style. Synthesis of pre-6a: step 5 (using Me3SiCl) is not comprehensible to me.
4. Molecular modelling: Please present a demonstrative graphic showing the binding of 3a in the manuscript (not only in the Supplementary) and discuss more clearly the input of the second boronic acid group in ortho position.
5. Conclusions: Important evidence: “Respective SAR analysis … indicates that most of the fluorinated derivatives maintain activity comparable to respective parent compounds. In turn, introduction of bulkier, lipophilic groups (I, CF3, SiMe3, OMe) has adverse effects and in general such compounds are not effective BLIs.” This is in discrepance to the final sentence “Thus, it seems that the concept involving introduction of the second boronic group to the structure can be considered as a promising tool for the future development of effective KPC/AmpC inhibitors.”. If only fluorine and hydrogen atoms are tolerated at the benzenediboronic acids, anf these have been entirely investigated here – where is space for future optimization??
6. Minor things: Line 73: “… well-known groups of competitive of β-lactamases inhibitors …”: delete the second “of”; line 101: “phenylboronic acids (7a−7d)”: change into 7a-7c; line 458: amino acids.
Reviewer 2 Report
Comments and Suggestions for Authors
The manuscript entitled “Aromatic diboronic acids as effective KPC/AmpC inhibitors ” by Joanna Krajewska et al. synthetize a series of boronic acids derivatives and evaluate the potency as β-lactamase inhibitors. It was found that ortho-phenylenediboronic acid 3a has highest activity against KPC-type carbapenemases, and exhibit strong synergy with the fractional inhibitory concentrations indices of 0.1-0.32. Overall, the manuscript is interesting. Aromatic diboronic acids is expected to be promising scaffold for the development of novel KPC/AmpC inhibitors. However, some areas need to be improved.
1. The authors need to confirm the 3a’s inhibition efficacy against purified KPC and AmpC using hydrolysis experiments and shows the hydrolysis curve diagram.
2. The covalent docking process of 3a with the catalytic Ser70 residue needs to be given in detail.
Comments on the Quality of English LanguageModerate editing of English language required
